# FROM OVERCONNECTIVITY TO SPARSITY: EMULATING SYNAPTIC PRUNING WITH LONG CONNECTIONS

## ABSTRACT

During brain development, an excess number of synapses are initially created, which are progressively eliminated through a process known as synaptic pruning. This procedure is activity-dependent, shaped by the brain's experiences. While creating an overabundance of synaptic connections only to later remove many might appear inefficient, research suggests that networks formed by this procedure demonstrate significant efficiency and robustness. Inspired by this biological process, we propose a neural network architecture utilizing long connections instead of traditional short residual connections. When long connection neural networks (LCNs) are trained with gradient descent, information is naturally "pushed" down to the first few layers, leading to a sparse network. Even more surprising is that this simple architectural modification leads to networks that exhibit behaviors similar to biological brain networks, namely: early overconnectivity to later sparsity, enhanced robustness to noise, efficiency in low-data settings and longer training times. Specifically, starting with a traditional neural network architecture with initial depth $d$ and $k$ connections, long connections are added from all layers to the last layer and summed up. During LCN training, 30-80% of the top layers become effective identity mappings as all relevant information is concentrated in the bottom layers. Pruning the top layers results in a refined network with a reduced depth $d'$ and final connections $k'$, achieving significant efficiencies without any loss in performance compared to residual baselines. We apply this architecture to various classification tasks and show that, in all experiments, the network converges to utilizing only a subset of the initially defined pre-training connections, and the amount of compression is dependent on the task complexity.

## 1 INTRODUCTION

Deep learning has achieved significant breakthroughs across diverse tasks and domains (Krizhevsky et al., 2012; LeCun et al., 2015; Brown et al., 2020); however, it still lacks the flexibility, robustness, and efficiency of biological networks. Modern models rely on deep architectures with billions of parameters, leading to high computational, storage, and energy costs. In contrast, biological brains are remarkably efficient, constrained by physical limitations and refined through evolution to develop low-power, fast-acting, effective networks (Bassett & Bullmore, 2006). These biological circuits achieve robust performance and rapid learning while maintaining low cost and power consumption. One key efficiency mechanism in the brain is synaptic pruning. During early development, an excess of synapses is formed and progressively eliminated through activity-dependent pruning (Sakai, 2020). Early studies (Peter R., 1979) measured synaptic density across different ages and found that it peaks around 1–2 years of age, followed by a decline to approximately $50\%$ by adulthood. Although creating an overabundance of connections only to remove many may seem inefficient, research indicates that this approach leads to networks exhibiting significant efficiency and robustness (Navlakha et al., 2015). Synaptic pruning is a gradual process that unfolds over the years and allows the brain to explore a large parameter space while learning and to identify key circuits by eliminating redundant connections. The result is a low-cost, sparse and efficient network that retains only the necessary pathways for information processing and routing, achieving strong performance and robustness.

Simply put, synaptic pruning is the evolutionary mechanism that *dynamically* optimizes the number of neural connections, effectively addressing the question: "How can we determine the essential parameters for efficient brain function during the learning process?" Contemporary deep learning

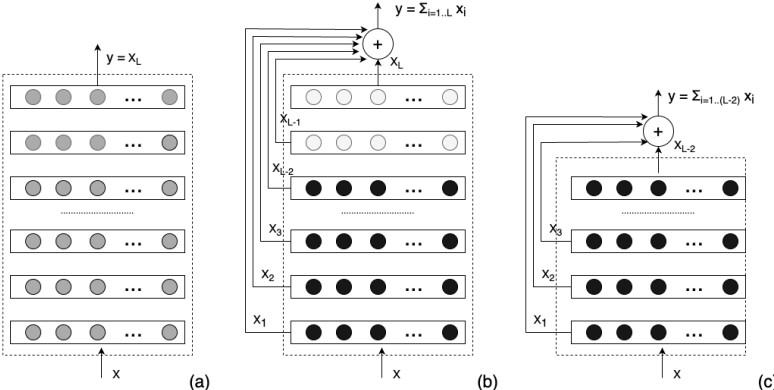

Figure 1: Main concept: (a) Start from a neural network with $L$ layers either randomly initialized or pretrained. (b) Add residual long connections from each layer to the ouput of the network and sum them (also remove any existing short residual connections - if any). During training/fine-tuning of the resulting LCN network the majority of the information (shown here as darker vs lighter circles) will naturally concentrate at the lower layers. (c) You may now safely remove the top (two in our example) layers during inference without any performance loss.

architectures aim for overparameterization; however, they lack a straightforward mechanism to identify and prune redundant connections. To address this, additional techniques are employed on top of the training process. $L_1$ and $L_2$ regularization (Ng, 2004) constrain network parameters to obtain small norms, often leading redundant parameters to have zero norm and be effectively pruned, especially in the case of $L_1$ regularization. Pruning techniques (Cheng et al., 2024) remove groups of redundant parameters without significantly affecting performance, usually after the full network is trained. Dynamic inference (Han et al., 2021) adjusts which parts of the network are used based on the input. However, these methods do not explicitly detect redundancy but instead bypass it using heuristic criteria layered on top of the training process.

In this work, we propose an *architectural modification* that replaces residual (short) connections with long connections, forming Long Connection Networks (LCNs). When long connections neural networks (LCNs) are trained with gradient descent, information is naturally "pushed" down to the first few layers, leading to a sparse network. This simple architectural modification leads to networks that exhibit behaviors similar to biological brain networks, namely: early overconnectivity to later sparsity, enhanced robustness to noise, efficiency in low-data settings and longer training times. Specifically, starting from an overparameterized network of depth $L$ and $k$ connections, LCNs refine it to depth $L'$ and connections $k'$, significantly accelerating inference and reducing memory usage without sacrificing performance. Our analysis shows that architectures with residual connections (He et al., 2016) or deep supervision (Lee et al., 2015) do not exhibit this behavior. We implement LCNs in fully connected and transformer-based architectures and find experimentally that they achieve similar or superior performance compared to residual baselines, while 30-80% of the top layers become effective identity mappings, as all relevant information is concentrated in the bottom layers. This approach is practical with current hardware and does not require specialized software. Finally, we highlight that this architecture can be used complementary with pruning or dynamic inference techniques.

## 2 LONG CONNECTION NETWORKS

The core idea behind LCNs [1] is to force each layer to produce discriminative features that are directly useful for prediction. In this manner, when the last layers are pruned, earlier layers can be used for prediction directly without the need for further fine-tuning. Concretely, we propose replacing the residual short connections with long connections, as described in Eq. 1 and shown in Figure 1 for a network of depth $L$[2]:

---

[1]Code for the paper is available here.

[2]We note that a classification head can be built on top of $y$.

$$x_i = f_i(x_{i-1}), \qquad y = \sum_{i=1}^{L} x_i \qquad (1)$$

In LCNs the output of each layer is directly connected to the output of the network, and thus is directly optimized by the objective function during gradient descent training. Furthermore, the number of possible shortcuts is equal to the number of layers, $L$. We find this simplification maintains the improved signal flow that shortcut connections provide, while also introducing the ability to determine the effective depth of the network, and thus perform layer pruning[3]. Compared to the similar idea of deep supervision, which introduces a separate loss function and classification head for each layer, LCNs are trained with a single loss function, aleviating the need for balancing the contribution of multiple objectives through hyper-parameter tuning, and achieving superior performance (see Appendix C). We also note that LCNs differ structurally from other models employing long connections, such as DenseNets Huang et al. (2017) and DenseFormer Pagliardini et al. (2024), which are residual networks variants. These two models connect each layer to all preceding layers whereas LCNs connect each layer only to the output, leading to a distinct structural design. [4].

## 2.1 MATH INTUITION

Inspired by the intuitive analysis by Bachlechner et al. (2021), we compare the Jacobian of a network, for a trivial one dimensional (1D) linear feedforward network with $L$ layers, where a weight $w$ is shared across all layers. We utilize three different architectures for this comparison: the classic feedforward architecture (MLP), the residual connections architecture, and LCNs[5]. The output $y$ of the network as a function of the input $x_0$ is shown in Eqs. 2- 4:

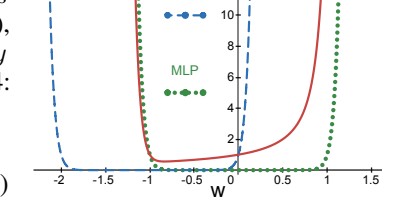

$$\text{MLP} \qquad \text{Residual} \qquad \text{LCN}$$

$$y = w^L x_0 \quad (2) \qquad y = (1+w)^L x_0 \quad (3) \qquad y = \frac{1 - w^{L+1}}{1 - w} x_0 \quad (4)$$

Figure 2: Jacobians of 1D MLP, residual and LCN architectures for shared weight and $L = 20$.

Their respective Jacobians for $L = 20$ are shown in Fig. 2. In the case of the MLP, we observe that for a large depth the Jacobian either vanishes or explodes as it traverses the layers of the network, depending on whether the value of $w$ is below or above 1, respectively. A similar behavior is observed in the residual architecture, but it is shifted such that for $w < 0$, the network experiences vanishing signal flow, while for $w > 0$, the Jacobian grows exponentially as a function of depth. The LCN architecture, however, exhibits a notable difference: for a substantial range of values $w$ around zero the Jacobian neither vanishes nor explodes as it propagates through the network. Given that most widely-used initialization schemes (e.g., He (He et al., 2015) and Xavier (Glorot & Bengio, 2010)) initialize weights around zero, LCNs offer a broader range of initial weight values that ensure consistent signal transmission. This expanded initialization window could enhance the effectiveness of the training process. Finally, we note that in Appendix G we deliver a broader theoretical analysis of LCN's training dynamics and discuss an interesting connection to layer-wise training.

## 2.2 A TOY EXAMPLE

To demonstrate the ability of the proposed architecture to compress information in early layers, we show a simple toy experiment involving a 1D linear feedforward network with three layers and weights $w_1$, $w_2$ and $w_3$ for each layer, respectively. The dataset comprises pairs drawn from the function $y = 2x$. We consider two architectures: the first employs residual (short) connections, while

---

[3]A careful reader may observe that long connections are a strict subset of the $2^L$ shortcut connections in residual networks. The exponential number of shortcuts in residual networks may be the reason that their effective depth is not easily determined.

[4]We mention more details about these models in Section 7

[5]A more analytic definition of the models and derivation of the equations is provided in Appendix A.

the second utilizes long connections (LCN). Given a synthetic dataset of $(x, y)$ pairs, the network should require only a single layer to successfully accomplish the task, i.e., $w_1 = 1$:

$$\hat{y}_1 = x_0 + x_1 = x + w_1 x = 2x, \tag{5}$$

effectively computing $y$ utilizing only the first layer ($x_1$ is the output of the first layer). We perform this simple training experiment multiple times ($N = 1000$), each time generating 1000 examples with values between $-10$ and $10$. The models are trained for 500 epochs and the weights are initalized with values around zero either uniformly or following a normal distribution. Fig. 3 illustrates the distribution of learned $w_1$ values for both architectures.

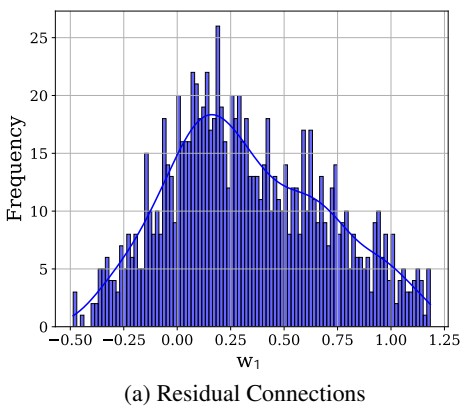
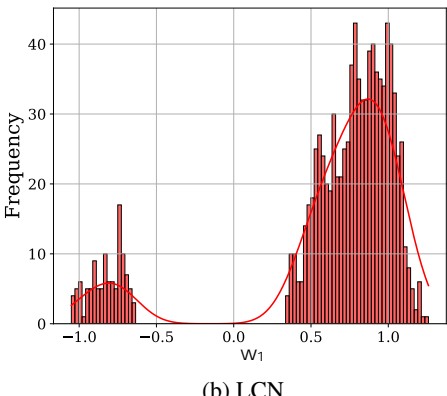

(a) Residual Connections  (b) LCN

Figure 3: Distribution of $w_1$, weight of the first hidden layer of a 1D linear feedforward network with three layers solving $y = 2x$, utilizing either: (a) residual connections or (b) long connections.

For residual architectures we observe a pretty wide distribution of $w_1$ values centered around $0.25$; indeed $y' = x + w_1 x + w_2 x + w_3 x \approx 2x$, for $w_1 = w_2 = w_3 = 0.25$ is a valid solution. Thus, in this example, *residual networks have the tendency to utilize all layers equally*, even if a sparse solution exists. LCNs, however, typically converge to solutions where $w_1$ is close to 1, allowing correct predictions from the first layer[6]. So in this simple example, *long connections in LCNs induce implicit depth regularization*, guiding the network toward sparse solutions.

## 3 PREDICTING WITH INTERMEDIATE LAYERS

In this section, we evaluate our proposed architecture across various modalities, datasets, and models, comparing its performance to the residual baselines.

### 3.1 EXPERIMENTAL SETUP

In our experiments, we utilize variants of the Transformer [Vaswani (2017)] and MLP-Mixer [Tol-stikhin et al. (2021)] architectures. For each input token, we compute a final output vector $y^t$ (where $t$ is the sequence index) by summing the output representations of all intermediate layers along with the input embedding, as shown in Eq. 1. To generate classification predictions, we either apply a pooling layer to these vectors for image classification or use the final representation of the `[CLS]` token for text classification. For a network of depth $L$, making predictions using $k$ intermediate layers involves computing $y_k^t$ for each token, which is the sum of intermediate representations up to layer $k$. This summed representation is then passed to the classification head. This approach yields $L + 1$ sub-networks, ranging from using only the input embedding (the first sub-network) to the full network (the last sub-network). For example, the network shown in Figure 1(c) would be the $L - 2$ subnetwork (utilizing layers 1 to $L - 2$), whereas the network in in Figure 1(b) would

---

[6]Note that initialization plays a crucial role, as $w_1$ sometimes converges near $-1$ for LCNs.

be the full network (all layers included). In baseline models with residual connections, the only difference is that when predicting with $k$ intermediate layers, we use the output representation of layer $k$ (that encapsulates all previous representations) as our $y_k^t$, without summing over layers. All other procedures remain the same. Additional experimental details and hyperparameters are provided in the Appendix.

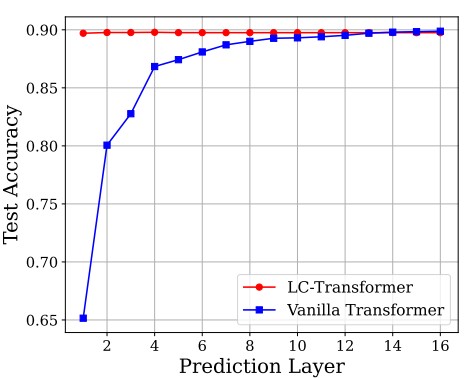 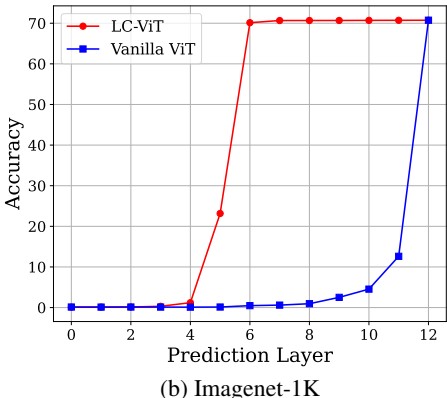

(a) Binary sentiment classification on Amazon reviews  (b) Imagenet-1K

Figure 4: Incorporating long connections into the Transformer architecture for text and image classification tasks.

## 3.2 TRANSFORMER WITH LONG CONNECTIONS

Transformers [Vaswani (2017)] are powerful architectures that utilize self-attention mechanisms, achieving state-of-the-art performance across various modalities and tasks (Dosovitskiy (2020), Brown et al. (2020)). Increasing their depth provides greater expressivity and yields competitive performance on complex tasks. However, for some simpler tasks the full utilization of all layers in a deep transformer architecture may be redundant. In Fig. 4a, we assess the effect of long connections in the Transformer architecture for binary sentiment classification using the Amazon reviews dataset (Zhang et al., 2015). For this purpose, we train a 16-layer vanilla Transformer with residual connections and a modified Transformer with long connections (LC-Transformer). To test the depth regularization capabilities of each architecture, we compute the classification performance of the embeddings of each layer $l = 1 \ldots 16$ without further model tuning. We observe that LC-Transformer can reach 90% accuracy, utilizing only 2 layers, while the vanilla transformer needs 13 layers[7].

To further demonstrate the effectiveness of the proposed long connections, we train a Vision Transformer (ViT) (Dosovitskiy, 2020) with long connections (LC-ViT) from scratch on the ILSVRC-2012 ImageNet-1K, following the training setup in the original paper. For both models we use 256 batch size due to memory constraints. LC-ViT converges at 700 epochs, while the vanilla ViT converges at 300 epochs[8]. As shown in Fig. 4b, LC-ViT reaches top performance at only 7 layers while the vanilla ViT needs all 12 layers to reach similar performance. So for both tasks and architectures, *LCNs can improve inference time and memory consumption without sacrificing performance*.

## 3.3 STABLE TRAINING OF DEEP LCN NETWORKS

One possible concern when replacing residual connections with long connections is that the gradient propagation suffers when increasing network depth. Next, we experimentally test the claim that *LCNs can compress information to the (same number of) first few layers during training, irrespective of the*

---

[7]Accuracy of just the input embeddings through the classification head here is 0.5 (2-classes).

[8]In this more challenging setting, we observe a trade-off between training and inference time, which is partially alleviated using a parameterization similar to DiracNets (Zagoruyko & Komodakis, 2017) for the MLP layers, specifically $\hat{W} = (I + W)$.

*network depth.* Specifically, we train a transformers with fixed layer width and increasing depth ($12$, $22$, and $40$ layers) on the IMDb binary sentiment classification dataset (Maas et al., 2011). In Fig. 5, we compare the behavior of the LC-Transformer to that of its vanilla counterpart as a function of layer performance. Overall, increasing the network depth does not lead to performance degradation for LCNs. Further, for all three experiments LCNs reach good performance utilizing only $2-3$ layers during inference. Of interest, is also the behavior of the residual network: performance progresses slowly with depth for $L = 12$ but more abruptly for $L = 22, 40$, a possible indication of overfitting.

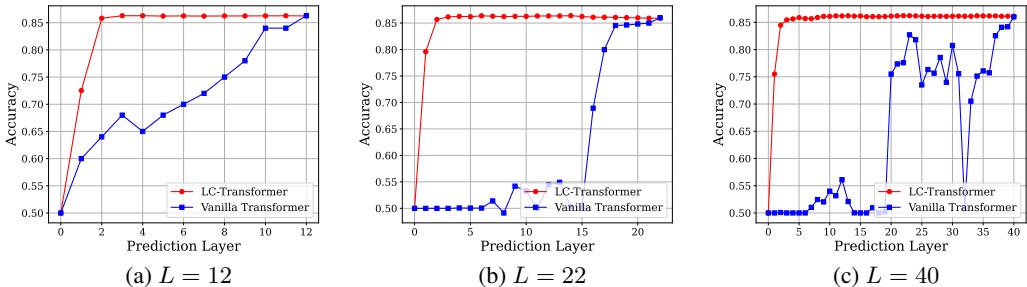

(a) $L = 12$        (b) $L = 22$        (c) $L = 40$

Figure 5: Performance of the intermediate layers of a Transformer with long vs residual connections as a function of network depth $L = 12, 22, 40$ (IMDb binary sentiment classification task).

### 3.4 THE EFFECT OF TASK DIFFICULTY

Intuitivelly, overparameterized networks trained on easier tasks should demonstrate higher levels of redundancy. Therefore, LCNs should converge to utilizing fewer layers as task difficulty decreases. To verify this, we use the number of classes as a proxy for task difficulty for image classification on the CIFAR-10 dataset (Krizhevsky, 2009). Specifically, we create subsets of 2, 5, and 10 classes, the assumption being that binary classification should be easier than 10-class classification. For this experiment we utilize MLP-Mixer (Tolstikhin et al., 2021) and train two variants, the original MLP-Mixer and the modified MLP-Mixer with long connections (LC-Mixer). Results are presented in Fig 6. We observe that indeed *LC-Mixer converges to solutions with larger effective depth, as the task "difficulty" increases.* Specifically, in this experiment, LCN needs 8, 10 and 12 layers for the 2, 5 and 10-class classification problem, respectively. In contrast, the vanilla MLP-mixer converges to solutions where the full depth of the network is utilized, irrespective of the task difficulty [9].

## 4 GENERALIZATION CAPABILITIES OF LCNS

An important difference between artificial deep neural networks and biological brains is their robustness to noisy input and sparse data. Biological brains excel in learning from limited data and are significantly more robust to noisy inputs. As discussed in Section 3.2, LCNs lead to sparser representations and required more training time than vanilla transformers on the ImageNet experiment. In this section, we experimentally test the generalization capabilities of LCNs to provide evidence for the question: *Does sparsity and longer training time of LCNs lead to "better" representations?*

### 4.1 ROBUSTNESS TO INPUT NOISE

Next, we present results assessing the robustness of LCNs versus vanilla transformer architectures to input noise. The experiments are performed with the LC-ViT and vanilla ViT architectures trained on ImageNet-1K (see Fig. 4b for baseline results with no noise). In this experiment, we inject increasing levels of additive Gaussian noise with standard deviation $\sigma = 0.1, 0.2, 0.4$, and salt-and-pepper noise with perentage of altered pixels $p = 1\%, 2\%, 10\%$. Results are shown in Fig 7(a) for Gaussian and

---

[9]Vanilla MLP-Mixer was trained for 300 epochs, while LC-Mixer for 420 epochs to reach the performance of the vanilla counterpart.

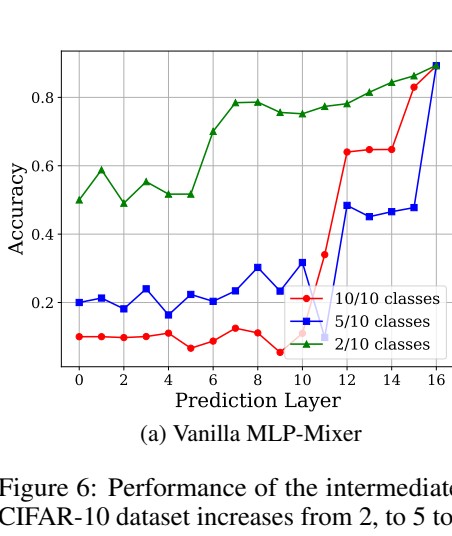
(a) Vanilla MLP-Mixer

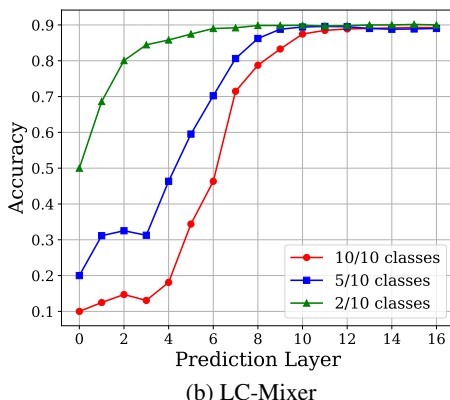
(b) LC-Mixer

Figure 6: Performance of the intermediate layers as the number of classes (and examples) in the CIFAR-10 dataset increases from 2, to 5 to 10 classes: (a) MLP vs (b) LCN.

(b) for salt-and-peper noise. We observe that *LCNs display improved robustness to noise*, and the performance gap in performance with the vanilla transformer increases as the noise levels increase.

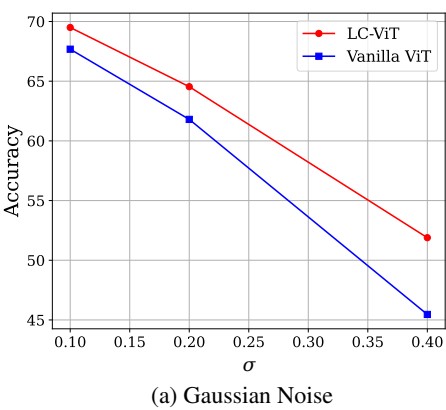
(a) Gaussian Noise

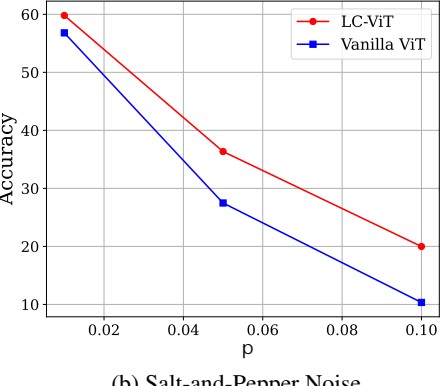
(b) Salt-and-Pepper Noise

Figure 7: Robustness of ViT with long connections (LC-ViT) and with residual connections (vanilla ViT) to additive Gaussian (left) and salt-and-pepper noise (right) on ImageNet-1K test set.

## 4.2 ROBUSTNESS TO DATA SPARSITY

Next, we experimentally compare the performance of residual and long connections architectures in low-data scenarios. For this purpose, we have created a subset of CIFAR-10 (Krizhevsky, 2009) by retaining only 100 samples per class, resulting in a total of 1000 examples. Using the same training settings and models as described in Section 3.4, we train both architectures for 150 epochs to assess how fast the training and test loss decrease, as a proxy for the generalization capabilities of each architecture. Results shown in Fig. 8, reveal that LCNs converge significantly faster for both

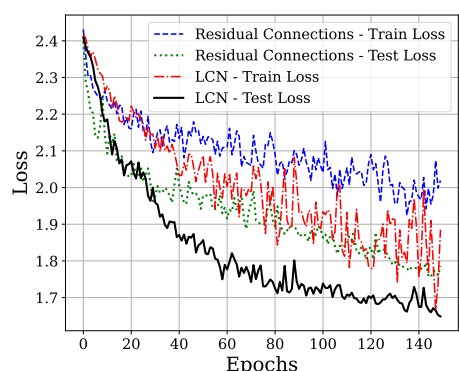

Figure 8: CIFAR-10 with 100 samples per class.

378
379
380
381
382
383
384
385
386
387
388
389
390
391
392
393
394
395
396
397
398
399
400
401
402
403
404
405
406
407
408
409
410
411
412
413
414
415
416
417
418
419
420
421
422
423
424
425
426
427
428
429
430
431

the training and test set, indicating that long con-
nections can be utilized in scenarios with limited
data.

## 5 EXAMPLE AND CLASS DIFFICULTY

An appealing feature of LCNs is their ability to determine the number of layers required for individual
examples in a dataset. Because each intermediate layer is connected directly to the final output, we
can evaluate each sub-network $i$ (comprising layers 1 to $i$) and check whether its prediction matches
that of the full network. We define the required number of layers for an example $e$ as $k$, where $k$
is the smallest sub-network whose prediction matches the full network's prediction. In Fig. 9a, we
show the histogram of the minimum number of layers $e$ required for each sample of the CIFAR-10
dataset. It is interesting to observe that for the vast majority of samples the correct decision is reached
by layer 5 for CIFAR-10, even if 12 layers are required to reach top performance (see Fig. 6b).
Similarly, one may compute the histogram of $e$ but separately for each class as shown in Fig. 9b.
This plot provides insights into the dataset's properties and could prove useful for data analysis and
exploration. An interesting future direction is to dynamically adjust the network's depth at inference
time on a per-sample basis, utilizing deeper layers only for more challenging samples and classes.
*Sample-dependent layer depth inference for LCNs* could further improve inference performance and
efficiency by an additional, e.g., 3-5 times on CIFAR-10 (similar to Dynamic Inference approaches
Han et al. (2021)).

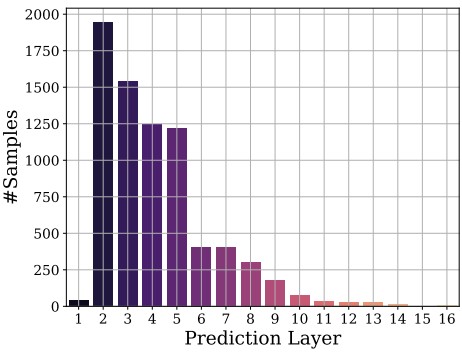

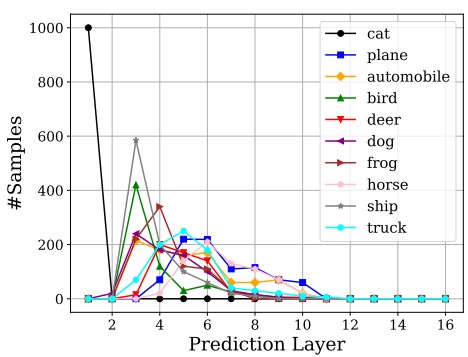

(a) Histogram of layer $e$ used per sample      (b) Histogram of layer $e$ used per sample and class

Figure 9: Histogram of minimum number of layers $e$ required for LCN to correctly classify a CIFAR-
10 sample computed: (a) per sample and (b) per class and sample.

## 6 SUMMARY OF RESULTS

From our experiments, we conclude that LCNs are able to "push" information down to the early neural
layers without performance degradation, resulting in a sparse high-performing network, effectively
revealing potential redundant layers and driving the pruning process. This pruning significantly
reduces memory requirements and accelerates inference. In all of the experiments we observed
that the converged depth between train and validation/test sets matched. Thus, pruning layer depth
is a meta-parameter determined directly on the validation set, eliminating the need for a separate
pruning procedure after training. Additionally, utilizing LCNs makes it straightforward to produce
and distribute differently sized variants of the same architecture with a single training run—for
example, distributing tiny, small, medium, and large versions of the model. In Table 1, we show the
performance of the pruned LCN models compared to their respective baselines. We also note that
utilizing the pruned LC-ViT instead of the vanilla base-ViT, we can reduce inference time from 13.9
miliseconds to 8.3 miliseconds (CPU). Finally, to further validate the general applicability of our
proposed architecture we conducted a BERT [Devlin (2018)] pre-training and fine-tuning experiment.
We direct the reader to Appendix F.

Table 1: Summary of experimental results (detailed in the previous sections) using pruned LCN models denoted with (p) vs full vanilla models denoted with (f).

| Models | Accuracy ↑ | #Parameters ↓ | #Inference Layers ↓ | Storage Size (MB) ↓ |
|---|---|---|---|---|
| Transformer on Amazon (f) | **90.05** ± 0.05 | 1.3M | 12 | 6 |
| **Transformer on Amazon (p)** | 90.02 ±0.07 | **0.75M** | **2** | **3** |
| MLP-Mixer on C10 (f) | 90.12 ±0.06 | 2.5M | 16 | 17.6 |
| **MLP-Mixer on C10 (p)** | **90.24** ± 0.05 | **1.8M** | **12** | **13.2** |
| ViT on ImageNet (f) | 70.74 ± 0.09 | 86M | 12 | 330 |
| **ViT on ImageNet (p)** | **70.76** ± 0.12 | **51M** | **7** | **195** |

## 7 BACKGROUND AND RELATED WORK

**Residual Connections:** Training deep neural networks with gradient descent becomes increasingly difficult as network depth increases. He et al. (2016) found that deeper convolutional neural networks (CNNs) not only suffer from declining generalization performance, often due to overfitting, but also experience a drop in training performance. This suggests that the challenge goes beyond overfitting and points to the inherent difficulty in optimizing deeper networks. To address this, they introduced residual connections (or identity mappings), proposing that learning residual functions relative to identity mappings simplifies optimization. These skip connections improve the training process and often enhance performance (Balduzzi et al. (2017), Orhan & Pitkow (2017), Zaeemzadeh et al. (2020), Li et al. (2018)). Veit et al. (2016) further argued that a residual network with $n$ layers can be viewed as a collection of $2^n$ paths of varying lengths. At each layer, the signal either skips the layer or passes through it, creating $2^n$ possible paths. Despite sharing weights, these paths function as an ensemble of networks, as confirmed by experiments. In contrast, a traditional deep feedforward network has only one path, so removing any random layer significantly degrades performance. Additionally, the authors showed that these paths are typically shallow, with backward gradients often vanishing after passing through only a small fraction of the total layers.

**Deep Supervision:** The authors of [Lee et al. (2015)] proposed adding complementary objectives to all intermediate layers, not just the final one. They argue that these intermediate objectives encourage hidden layers to learn more discriminative representations, improving the overall classification task. The intuition is that more discriminative features lead to a better-trained classifier. In their approach, each intermediate objective $i$ is a loss function that captures the classification error of an SVM trained on the output features of layer $i$. The overall loss is the sum of the intermediate and final objectives, and results show that this improves final classification performance.

**Structure Learning (Depth Optimization):** Depth optimization focuses on learning the optimal network depth while training. Alturki et al. (2023) introduced "weight relevance of a layer," a metric that measures each layer's importance in the classification task. Using this metric, they propose dynamically adjusting the network's depth by removing irrelevant layers. Similarly, Cortes et al. (2017) presents an algorithm for learning both width and depth during training. Starting with a simple linear model, layers are added competitively based on a trade-off between complexity and performance, resulting in a comprehensive structural learning algorithm, backed by strong theoretical analysis.

**Long Connections in the Literature:** Brain networks combine short and long connections [Bassett & Bullmore (2006)], where short connections form dense sub-network hubs, and long connections sparsely link these hubs. Typically, short connections are more numerous and have stronger synaptic weights [Muldoon et al. (2016)]. In [Betzel & Bassett (2018)], the authors show that short connections more efficiently route information across brain areas and sub-networks. Removing short connections has a larger impact on network properties like average path length. In contrast, long connections are key for functional diversity, offering unique inputs and novel targets for outputs across sub-networks. DenseNet [Huang et al. (2017)] incorporates a form of long connections within CNN residual blocks, connecting every two layers within a block. This design enables feature reuse and efficient signal propagation, mitigating vanishing gradients while reducing parameters and computation without sacrificing performance. More recently, DenseNet's concept was applied to transformers in DenseFormer [Pagliardini et al. (2024)], where each layer receives a weighted sum of outputs from all preceding layers. This approach improves training efficiency, speeds up inference, and reduces memory requirements. The learned weights show strong reuse of distant layers' outputs, ensuring efficient information flow in the network.

## 8 DISCUSSION

In this work, we introduced Long Connection Networks (LCNs), where, starting with a traditional neural network architecture, long connections are added from all layers to the last layer and summed up. During training with gradient descent, these networks mimic synaptic pruning (albeit in a layer-wise fashion) by beginning with

an overconnected network and gradually identifying the key connections needed for the task. LCNs require longer training but ultimately produce more robust, efficient, and sparse networks, similar to the development of biological neural networks. Additionally, their training dynamics align with those of layer-wise training, further resembling processes observed in cognitive neural development in the prefrontal cortex [DeFelipe (2011)]. By training fully connected and transformer-based architectures with long connections, information is naturally "pushed" down and concentrated in the bottom layers, rendering the top layers into effective identity mappings, leading to significant inference time efficiencies. We also found that the amount of compression that LCNs achieve is dependent on the number of classes, a simple proxy for task compexity in classification settings.

In our experiments, we observed a trade-off between training and inference cost when choosing between short residual connections and long connections. It is possible that LCN's inherent sparsity and longer training time is what makes them more robust to noise and more efficient in low-data settings. Preliminary experiments further strenghten this belief: as the representations learned by earlier layers become more discriminative focusing on the task at hand, LCNs can still effectively transfer their knowledge to downstream tasks (Appendix D).

Future research could expand LCNs to self-supervised and multi-task settings, leveraging the pre-training and fine-tuning paradigm. LCNs also hold promise for generative tasks, reducing inference costs and energy consumption. Additionally, developing inference-time algorithms that dynamically adjust the number of layers per sample for optimal performance and efficiency is an intriguing direction for future work. Last but not least, LCNs are only one possible long-connection architecture out of the many that are worth investigating further.

## 9 LIMITATIONS

Due to resource constraints, our proposed architecture was evaluated solely on classification tasks; however, it demonstrated robust and promising performance across various modalities, datasets, and state-of-the-art architectures within this scope. To fully assess its potential and limitations, further testing on a broader range of tasks is essential. Additionally, applying our method in self-supervised and multi-task learning settings, such as training large-scale language models or multimodal models, represents a significant and exciting avenue for future research.

Another limitation is the increased training time observed with LCNs compared to traditional architectures with residual connections. While we partially addressed this issue by employing parameterizations similar to DiracNets, a more comprehensive solution to reduce training time remains an open question. Of course this could be both a blessing and a curse, as longer training times might lead to learning better representations. In any case, further research into training schedules and initialization schemes is needed to resolve this trade-off.

## 10 BROADER IMPACT

Our work contributes to the development of more efficient and robust neural network architectures by drawing inspiration from biological processes. By enabling networks to identify and prune redundant layers, we aim to reduce computational, memory, and energy requirements during inference, which will have a significant impact with broader adoption of AI technology. Conceptually, this line of research could lead to network architectures with inherent System 1 and System 2 capabilities (Kahneman, 2011), where networks adaptively use fewer layers—analogous to fast thinking—for easier tasks, and engage more layers—resembling slow thinking—for more complex tasks.

However, as with any advancement in AI, there is a potential for misuse. More efficient models could be leveraged to deploy AI systems more broadly, including in areas with insufficient oversight or in applications that may infringe on privacy or other ethical considerations.

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

# A  1-D LINEAR JACOBIAN DERIVATION

## A.1  MLP

$$y = w_L w_{L-1} \ldots w_1 w_0 x = w^L x_0 \qquad (6)$$

## A.2  RESIDUAL CONNECTIONS ($x_{l+1} = f_{l+1}(x_l) + x_l = w_{l+1}x_l + x_l$)

$$y = \sum_{k=0}^{L} \binom{L}{k} w^k = (1 + w)^L x_0, \qquad (7)$$

from the binomial formula.

## A.3  LONG CONNECTIONS

$$y = (x_0 + x_1 + \cdots + x_{L-1}) = (x_0 + wx_0 + w^2 x_0 + \ldots) = \frac{1 - w^{L+1}}{1 - w} x_0 \qquad (8)$$

# B  EXPERIMENTAL DETAILS

**Amazon Polarity - Transformer** We are using Transformers with pre-norm, hidden dimension equal to 64, MLP dimension equal to 512 and the number of heads is 4. We are using the AdamW optimizer Loshchilov (2017) with a maximum learning rate equal to 0.001. The number of layers is 12.

**IMDb - Transformer** We are using Transformers with post-norm, hidden dimension equal to 64, MLP dimension equal to 512 and the number of heads is 4. We are using the AdamW optimizer Loshchilov (2017) with a maximum learning rate equal to 0.001. The number of layers changes as depicted in the respective Figures.

**CIFAR-10 - MLP Mixer** The MLP Mixers have 16 layers with a hidden size of 128. The patch size is 4 (the input is 32x32, 3 channels). The MLP dimension $D_C$ is 512, while $D_S$ is 64. We are using the AdamW optimizer Loshchilov (2017) with a maximum learning rate of 0.001 and a Cosine Scheduler with Warmup.

# C  DEEP SUPERVISION DOES NOT EXHIBIT THE SAME BEHAVIOR

One could argue that since the deep supervision approach introduces losses for intermediate layers, it may potentially converge to a similar behavior (as this approach intuitively encourages all layers to produce discrminative representations). To evaluate the behavior of deep supervision-like methods, we trained two variants on the CIFAR-10 task using the MLP-Mixer architecture: one with a shared classification head across all intermediate layers but with a separate auxiliary loss for each layer, and another with a unique classification head (a linear classifier) for each layer. Based on the results shown in Figure 10, we conclude that deep supervision objectives fail to demonstrate the expected behavior. Specifically, the sub-networks in these models behave similarly to those in regular residual architectures.

# D  IMPACT ON EARLY LAYERS' GENERALIZATION PROPERTIES

It is well established that early layers in deep neural networks tend to learn more general and transferable features, while later layers specialize in task-specific representations Yosinski et al. (2014). Connecting all layers to the output and "pushing" down information to the bottom layers could potentially lead to a model that is more specialized for the task and less transferable to downstream tasks. To investigate this, we fine-tuned both the vanilla Vision Transformer (ViT) and the long connections ViT (LC-ViT) that we trained on ImageNet, on CIFAR-10. As shown in Figure 11, our results indicate that this is not the case. Both models achieve similar final accuracy, and they converge in roughly the same number of epochs (approximately 20 epochs). For LC-ViT, we evaluated both the full and pruned versions (as shown in the dashed lines). It is important to note that we did not optimize for top performance in this experiment, such as by using higher resolutions or additional techniques from the original paper. The primary goal of this experiment was to demonstrate that the proposed architecture does not suffer from a loss of transferability.

# E  VISION TRANSFORMER ON IMAGENET TRAINING CURVES

Here, we present the predictions of the sub-networks at various epochs during training for both vanilla and LC-ViT. Two key observations emerge. First, the residual ViT architecture trains faster, something that is

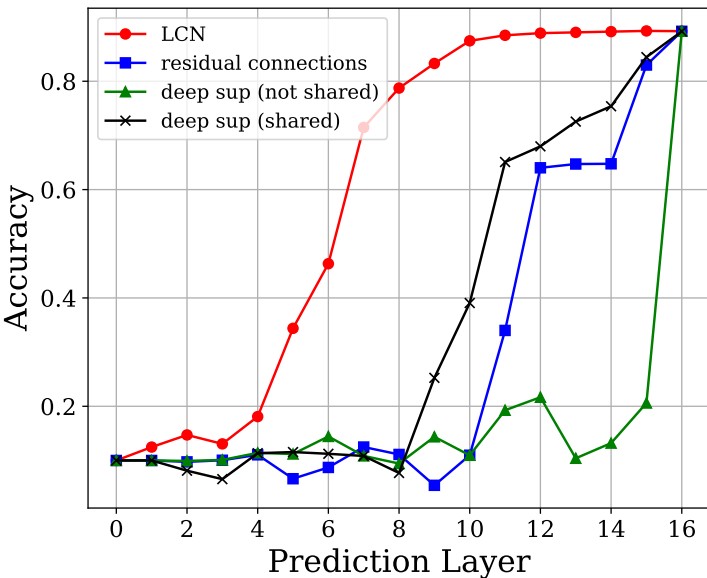

Figure 10: Deep supervision objectives fail to exhibit behavior similar to long connections architectures, even though their objectives are somewhat similar.

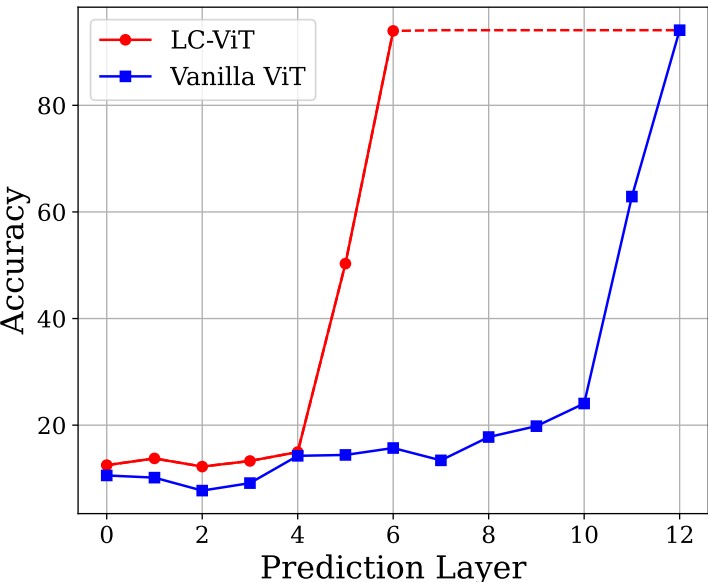

Figure 11: Fine-Tuning the previously trained ViTs on CIFAR-10. The long connections architecture transfers to downstream tasks as effectively as the vanilla counterpart.

depicted also in the number of epochs its model is trained. Second, LC-ViT provides an indication of the required number of layers early in training, with the converged depth becoming clearly apparent as early as epoch 30 out of 700. This behavior could be leveraged to formalize an algorithm for early pruning of redundant layers, thus effectively speeding up training.

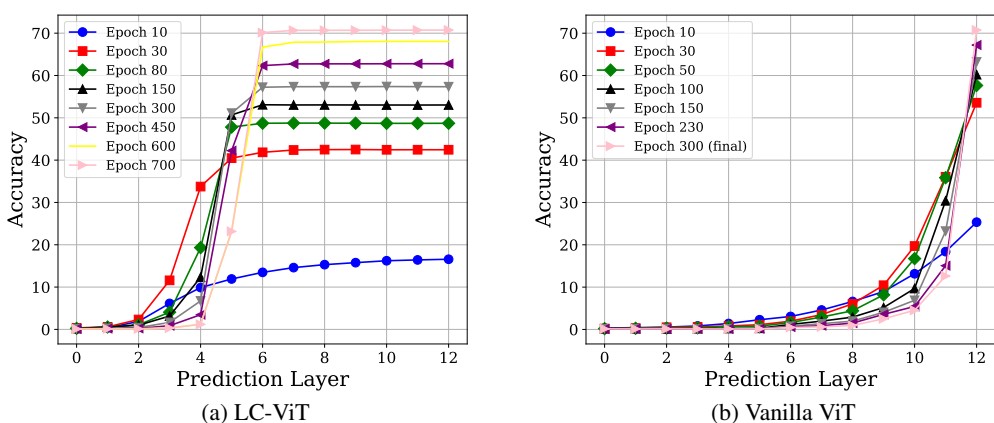

(a) LC-ViT

(b) Vanilla ViT

Figure 12: Behaviour of ViT sub-networks of residual and long connections architectures during training on ImageNet 1.3M

# F    MASKED LANGUAGE MODELING AND TRANSFER LEARNING WITH LONG CONNECTIONS

Since in the main paper we focused on classification tasks, we conduct here an experiment on BERT pre-training and fine-tuning to further demonstrate the broad applicability of the LCN architecture. Due to limited resources, we train vanilla BERT for one epoch, while LC-BERT with our proposed long connections, for two epochs to match the performance of the vanilla counterpart (again requiring more training time). We utilize the same pre-training corpus as the original model (BooksCorpus Zhu (2015) + English Wikipedia). After training, we fine-tune the models using the original BERT Devlin (2018) setup on three datasets from the GLUE benchmark [Wang et al. (2018)], namely SST-2 [Socher et al. (2013)], QQP and QNLI [Rajpurkar (2016)]. In Figure 13, we observe that the LCN variant converges to significantly fewer layers ($\widetilde{7}5\%$ less) than the vanilla baseline, while maintaining on-par performance. This introduces intriguing possibilities, such as pre-training LLMs with long connections and adaptively leveraging only a subset of the full model by finetuning on downstream task.

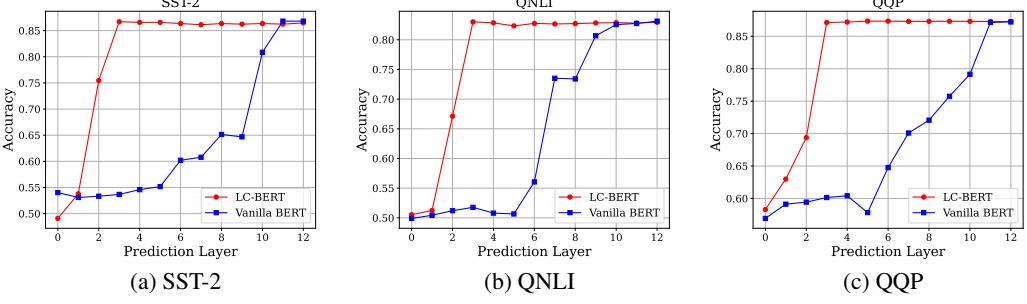

(a) SST-2

(b) QNLI

(c) QQP

Figure 13: Downstream performance of LC-BERT vs vanilla BERT, pre-trained for one epoch on original BERT's pretraining corpus.

# G    ANALYSIS OF LCN TRAINING DYNAMICS

## G.1    INTRODUCTION

For simplicity we consider linear neural networks of depth $d$ and derive the equations for the 1-dimensional case (they can be expanded to $N$-dimensional inputs).

**Notation** $x_i$ is the output of layer $i$ and $w_i$ is the weight of layer $i$ (the weight used to construct $x_i$). Then, $x_0$ is the input (after the embedding) and $y$ is the output.

## G.2 FFNs AND LCNs

FFN forward pass

$$y(= x_d) = \prod_{i=1}^{d} w_i x_0 \tag{9}$$

FFN backward pass for weight $i$

$$\frac{\partial y}{\partial w_i} = \frac{\partial y}{\partial x_i} \frac{\partial x_i}{\partial w_i} \tag{10}$$

$$\frac{\partial y}{\partial w_i} = \underbrace{\left(\prod_{k=i+1}^{d} w_k\right)}_{\text{"after" path}} \underbrace{\left(\prod_{m=1}^{i-1} w_m\right)}_{\text{"before" path}} x_0 \tag{11}$$

LCN forward pass

$$y = x_0 + \sum_{i=1}^{d} x_i = x_0 + \sum_{i=1}^{d} \prod_{j=1}^{i} w_j x_0 \tag{12}$$

LCN backward pass for weight $i$

$$\frac{\partial y}{\partial w_i} = \frac{\partial y}{\partial x_i} \frac{\partial x_i}{\partial w_i} = \left(1 + \sum_{k=i+1}^{d} \frac{\partial x_k}{\partial x_i}\right) x_{i-1} \tag{13}$$

$$\frac{\partial y}{\partial w_i} = \underbrace{\left(1 + \sum_{j=i+1}^{d} \prod_{k=i+1}^{j} w_k\right)}_{\text{"after" path}} \underbrace{\left(\prod_{m=1}^{i-1} w_m\right)}_{\text{"before" path}} x_0 \tag{14}$$

**Observations** We see that the gradient of $w_i$ can be decomposed in two terms, which we coin as "before" and "after" path. These two paths play a major role in the gradient of each weight as well as the general training dynamics of the network. LCNs have "before" path equivalent to FFNs but different "after" path due to the long connections. We will argue that this is the reason why LCNs behave in the way we saw in the experiments.

If we consider the extreme case where all weights are initialized to zero, we see that in the case of LCN in the first training batch only the gradient of the first layer's weight will be non-zero. Moreover, for a weight $i$ to have a non-zero gradient, all the previous weights need first to be trained (non-zero gradient). Also, in the forward pass, only the trained subnetworks will have a contribution to the output addition, effectively solving the task with only those. So there is a gradual hierarchical training from early layers to later ones. If the first $k$ layers achieve low task error, effectively minimizing the loss, then the gradient flow to all layers will be small effectively rendering the last $d - k$ layers "under-trained".[10]

Although initializing all weights to zero is unrealistic, most popular initialization methods today initialize weights with small norms, close to zero. As a result, the earlier analysis remains applicable: early LCN layers tend to have significantly larger gradients initially due to the small weight norms, causing them to train faster and contribute more to solving the task early on. Gradually, as the early layers are trained, the deeper layers begin to train as well, depending on the complexity of the task. For instance, if the early layers successfully minimize the loss function and solve the task, the gradients for the deeper layers will remain small. On the contrary, in FFNs all weights have gradients with similar magnitudes initially (since gradient of weight $i$ contains the product of all other weights and the input) and so they are all trained in a similar rate. Also notice that as the depth increases these products grow smaller and smaller, making the gradients also small and the training slow (vanishing gradients).

---

[10]Note that this depends very much on the initialization. For example, in Fig 3, where a wide range of values were used for initialization, the LCN does not always converge a solution that only uses the first layer.

### G.3 FORWARD VS BACKWARD PATHS (WHY RESNETS DON'T GET THE JOB DONE)

For residual networks, the functional form induced by adding the identity at each layer introduces an exponential number of paths and thus makes the backward pass more complicated. For illustrative purposes, we will consider a simplified version of residual networks with a smaller number of paths. Specifically we will consider a residual network architecture that can be thought of as a reversed or forward LCN (rLCN) where each layer is receiving as input the output of the previous layer plus the input embedding $x_0$. In essence, in LCNs we draw a residual connection from each layer to the last layer, while in rLCNs we add residual connections from the input embedding $x_0$ to all layers. We argue that this "backward" vs "forward" residual connections is what makes a difference to the derivative flow dynamics during training. Specifically for rLCNs:

rLCN forward pass

$$y = x_d \tag{15}$$
$$x_i = w_i x_{i-1} + x_0 \tag{16}$$

rLCN backward pass for weight $i$

$$\frac{\partial y}{\partial w_i} = \frac{\partial y}{\partial x_i} \frac{\partial x_i}{\partial w_i} \tag{17}$$

$$\frac{\partial y}{\partial w_i} = \underbrace{\left(\prod_{k=i+1}^{j} w_k\right)}_{\text{"after" path}} \underbrace{\left(1 + \sum_{m=1}^{i-1} \prod_{r=m}^{i-1} w_r\right)}_{\text{"before" path}} x_0 \tag{18}$$

This simple modification (that can be thought as a proxy or simplified traditional residual network architecture) changes only the "before" path of the network (compared to FFNs). Although, the modification of the "before" path adds significant robustness to the training process (by ameliorating the vanishing gradients effect) it does little to encourage earlier layers to get trained first, i.e., the derivative flow dynamics are not much different than in the FFN case because the "after" term is identical to FFNs. The analysis can be extended to other residual architectures, as well as, to novel architectures that combine "forward" and "backward" residual connections.

To summarize, in FFNs and Residual Networks all layers are trained in a similar rate; in the case of FFNs the rate is relatively slow especially as the depth increases (vanishing gradients), whereas in the residual connections case training is efficient.

So based on this analysis we expect that in terms of training speed earler layers will be trained slowest for FFNs, followed by residual networks (that might show a slight improvement due to the "before" term), while LCNs train their earlier networks fastest (due to the "after" term).

### G.4 IN PRACTICE

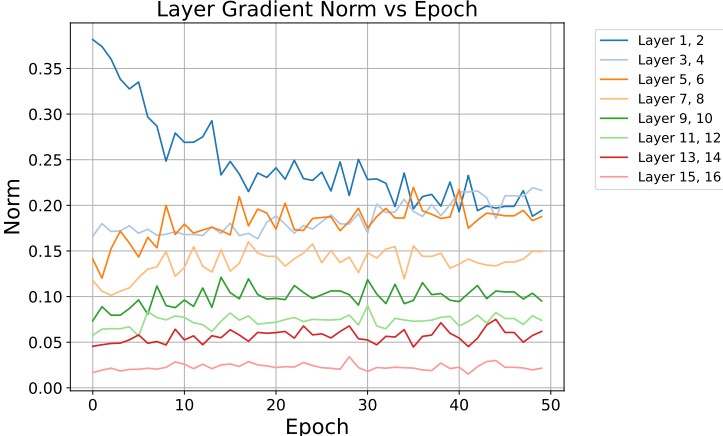

Figure 14: Layer Gradient Norms (normalized to sum up to 1) during training of LC-Mixer on CIFAR-10.

We validate the theoretical claims of the previous section by monitoring the gradient dynamics of LCNs during training. Specifically, we track the gradient norms of LC-Mixer while training on CIFAR-10. For clarity we show 50 epochs of training and we form blocks by summing every 2 layers' gradient norms. Additionally, the gradient norms are normalized to 1 at each epoch, allowing the plot to reflect the percentage contribution of each block's gradient norm to the total. Figure 14 validates our analysis; we observe that bottom layers have larger gradient norms than deeper layers throughout the training [11].

Note: Our analysis is further verified by the recent publication [Chen et al. (2023)], e.g., see Figure 2. Compare the results for FFNs, ResNets shown in the paper with the figure above[12].

# H    CONNECTION TO GREEDY LAYER-WISE TRAINING

Greedy layer-wise training [Hinton et al. (2006), Bengio et al. (2006)] was a popular method for training deep neural networks in a sequential manner, inspired by cognitive neural modeling. Based on the conducted mathematical analysis on LCN's dynamics, we argue that LCNs share similarities with the method and can be viewed as a "one-shot" version of it. In layer-wise training, the process is sequential: starting from the first layer, each layer is trained individually while the other layers remain frozen, with training progressing from first to last layer. Our analysis of LCN's dynamics shows that a similar sequential layer-wise training process occurs in LCNs, where the early layers train first, followed by the deeper layers. The key difference is that in LCNs, this layer-wise training happens naturally, without the need to freeze layer gradients or introduce hyperparameters like the number of epochs for each layer's training. Interestingly, this behavior is inherently driven by the architecture of LCNs (the structure imposed by the long connections), rather than being externally imposed.

---

[11]We note that each Mixer layer comprises four fully connected layers. To calculate the gradient norm for a Mixer layer, we take the average of the gradient norms of these four layers.

[12]Also in Figure 1 Chen et al. (2023) we observe that deeper layers in both FFNs and ResNets have larger gradients. In contrast, as demonstrated both theoretically and empirically, LCNs show the opposite behavior.

