# OpenReview forum: "From Overconnectivity to Sparsity: Emulating Synaptic Pruning with Long Connections"
_ICLR.cc/2025/Conference — Submitted to ICLR 2025_

### Official Review · Reviewer_Fzkp · 2024-10-25

**Soundness:** 3
**Presentation:** 3
**Contribution:** 3
**Rating:** 6
**Confidence:** 4

**Summary:**

The paper explores a new neural network architecture inspired by biological brain processes, specifically synaptic pruning. The paper proposes Long-Connected Neural Networks (LCNs) as an alternative to traditional deep learning architectures that primarily use short residual connections.
This simple architectural modification leads to networks that exhibit behaviors similar to biological brain networks, namely: early overconnectivity to later sparsity, enhanced robustness to noise, efficiency in low-data settings and longer training times.

**Strengths:**

- Good Writing. The paper clearly explains the motivation for the proposed work, drawing a strong connection between biological processes and deep learning archutectures.
- The paper proposes a novel architecture. By pushing information to the lower layers and pruning upper layers, the architecture addresses the common issue of overparameterization in deep learning models. It is interesting.
- The paper demonstrates that LCNs lead to sparser networks and improved efficiency. This can be particularly valuable in real-world applications where computational resources and memory are limited.

**Weaknesses:**

- The paper provides some mathematical intuition, but a deeper theoretical analysis of why this architecture works so well (e.g., in terms of information bottleneck theory or gradient dynamics) could strengthen its claims.
- Across the experiments, I cannot find the comparative experiments with state-of-the-art pruning techniques.

**Questions:**

- Please explain why your method can have effect.
- Please compare with some state-of-the-art pruning techniques.

---

> ### Author Response · Authors · 2024-11-26
> **Reply to Reviewer Fzkp**
>
> We thank the Reviewer for taking the time to examine our work and for providing valuable insights and comments. We will now address the questions raised:
>
> 1. “Please explain why your method can have effect.”
>
> We present a detailed mathematical analysis of the training dynamics of our method in Appendix G. Our analysis of the gradient flow dynamics in LCNs supports our empirical observations. In Appendix G4 we also include visualizations of the gradient norms of LCNs during training, which are also aligned with our analysis and observations. Essentially, long connections impose a structure that modifies the gradient dynamics of the network (compared to FFN or residual) and supports "layerwise training".
>
> 2. “Please compare with some state-of-the-art pruning techniques.”
>
> We provide a reference to a recent work (ICLR ‘24) [1] that focuses on Vision Transformers pruning. We refer to Tables 4 and 5, where the authors report results on ImageNet using ViT-Base, allowing for direct comparison. In Table 4, they provide the FLOPs and accuracy drop with their method, while in Table 5, they report similar metrics for ToME [2], which is not a pruning technique but rather a method for reducing the number of tokens and, consequently, the model's FLOPs. With our method, we achieve 10.30 FLOPs (G), outperforming both approaches without sacrificing accuracy. We provide the table below based on ViT-Base on ImageNet,  for comparison (the numbers of the two methods are extracted from Table 5 in [1]):
>
> | **Method** | **Acc. Drop(%)** | **FLOPs(G)** |
> |--------------|--------------|--------------|
> | OPTIN [1] |  0.71 | 11.45  |
> | ToME [2]    |  1.88   | 11.50 |
> | LCN (ours)     | 0.0 | 10.30 |
>
> [1] https://arxiv.org/abs/2403.17921, [2] https://arxiv.org/abs/2210.09461

---

> > ### Comment · Reviewer_Fzkp · 2024-11-27
> > **Reply**
> >
> > Thank the authors for providing additional experiment results by referring to the Reference [1].
> > First, your copy was not right, please check Tables 4 and 5 in the [1].
> > Second, I don't think one comparison is enough to show your method's superiority.
> > Lastly, I will wait for other reviewers' replies to determine the last score.

---

> ### Author Response · Authors · 2024-11-28
> **Reply to Reviewer Fzkp**
>
> Thank you for your follow-up comments. We acknowledge the concerns you have raised:
>  - We apologize for the discrepancy in our copying from (https://arxiv.org/pdf/2403.17921) in our initial reply. We corrected our table.
>
>
>  -  We understand the Reviewer's concern that a single comparison point may be insufficient to demonstrate the method's superiority.  To address this, we expand our comparison by referring to Figure 6 of [4], which presents several pruning methods for BERT-Base. Notably, our method, which removes entire layers, results in a 0.25 relative FLOPs with no performance degradation, outperforming all methods shown in the figure. Additionally, since our approach can be viewed as a natural compression method, we include results from [1], a 2024 survey on general transformer compression. In Table 4, the authors compare works that compress BERT-Base using knowledge distillation. Since we provide results (Appendix F) only for 3 datasets of the GLUE Benchmark (SST-2, QNLI, QQP) we provide the original papers' references which contain the analytic results on GLUE. We observe that:
>
> From Table 1 of [3], TinyBERT with 14.5M parameters exhibits an accuracy drop for 2 out of 3 datasets (from ~1% to ~3%) depending on the task. TinyBERT with 66.7M parameters performance is almost on par with the full BERT-Base. For DistilBERT (66M parameters), from Table [1] of [2] we see that there is an accuracy drop of ~1-2.5% on the 3 datasets. Our model converges on 3 layers (45-46M parameters) without accuracy drop compared to the residual baseline (full model), striking a balance between efficiency and performance.
>
>  [1] https://arxiv.org/pdf/2402.05964
>
>  [2] DistilBERT: https://arxiv.org/pdf/1910.01108 (Table 1)
>
>  [3] TinyBERT: https://arxiv.org/pdf/1909.10351 (Table 1)
>
>  [4] https://arxiv.org/pdf/2204.09656
>
>  We hope that this provides a more complete picture of our method's performance relative to existing pruning/compression approaches.
>
> Finally, we would like to emphasize that our work is not solely focused on pruning; instead, it ultimately introduces a new architecture, Long Connection Networks, which you also highlight in the strengths section of your initial review. So, while pruning is certainly one valuable outcome, we believe that our work also makes significant contributions to neural network design, theoretical understanding, and practical implementation, all of which are central to the paper. The architecture seems to naturally induce several desirable properties (namely, natural sparsification, enhanced robustness to input noise, efficiency in low-data settings) that could be valuable for making neural networks more efficient, robust, and biologically plausible.
>
>
> We appreciate your thorough review process as well as your feedback and will await the consolidated feedback from all reviewers. We are committed to addressing all concerns raised to strengthen the paper's contribution to the field.

---

> > ### Comment · Reviewer_Fzkp · 2024-12-03
> > **Reply**
> >
> > After reviewing all your responses and the feedback from other reviewers, I have decided to maintain my score. However, I still have concerns regarding the simplicity of the evaluation and comparison.

---

### Official Review · Reviewer_Zpq5 · 2024-11-02

**Soundness:** 2
**Presentation:** 3
**Contribution:** 2
**Rating:** 5
**Confidence:** 4

**Summary:**

The authors propose a method for encouraging more task-informative representations in the lower layers of ANNs and thereby improving efficiency.

**Strengths:**

- the text is generally well written

**Weaknesses:**

- I did not feel that the comparison to similar models was adequately made. There is a related work section which many relevant models are correctly brought up, but 1. there is no explicit description of how LCNs are different from these models and 2. apart from residual connections, none of these models are quantitatively compared to this previous work. One is left to ponder whether LCNs makes conceptual/quantitative *advances* in the field.

- the link to biology and the brain is interesting, but after the initial abstract it is hardly addressed. The authors report longer training times as more akin to 'biological brain networks', but I'm not sure which result even supports this (in Fig 8 it trains faster?). That LCNs handle noise better is interesting, but an intuitive explanation/link to previous literature as to why this occurs would be very helpful in its interpretation.

- It appears from the figures that all results are derived using a single random initilisation seed? If so, I would strongly recommend re-running these analyses over several seeds, otherwise it is very difficult to draw any strong conclusions.

- Some of the segways between sections could be smoother. For example short introductions to sections 3.1/7 would help. Some figures could also be clearer. For example a few instances where the axis labels were confusing. E.g. what are the axis labels in Fig 2? I presume w and the error gradient with respect to w? in fig 3 is weight of hidden1 = w1, and wouldn't it be interesting to look at w2/w3 as well to check if they are sparser for LCN, and maybe absolute values instead of raw values to better show sparsity? In Fig 7 I would prefer an explicit mathematical term relating to an equation than 'std' and 'noise_level'. In section 3.1 a schematic (even if for the appendix) would be helpful to the reader

**Questions:**

- line 54: 'Contemporary deep learning architectures aim for overparameterization' - what does this mean?
- is the residual network given by equation (3) a specific type of residual network? It is introduced there rather abruptly for a non-expert reader
- for the toy example the models are trained for 500 epochs (1000 examples per epoch). This seems like a lot for such a simple problem?
- line 259: 'possible indication of overfitting' - is Fig 5 showing the training or validation accuracy?

- typos:
line 210: brackets around citation
line 326: -> architectures
line 411: -> directly
paragraph 460: brackets around (multiple) citations
line 492: -> effectively

---

> ### Author Response · Authors · 2024-11-26
> **Reply to Reviewer Zpq5 (1/2)**
>
> We thank the Reviewer for taking the time to examine our work and for providing valuable insights and comments. We will now address the questions raised:
>
> 1. ‘I did not feel that the comparison to similar models was adequately made. There is a related work section which many relevant models are correctly brought up, but 1. there is no explicit description of how LCNs are different from these models and 2. apart from residual connections, none of these models are quantitatively compared to this previous work. One is left to ponder whether LCNs makes conceptual/quantitative advances in the field.”
>
> - Qualitative comparison of LCNs and Related models
>
> We revised the paper to further highlight our contribution in relation to the existing literature (Section 2 line 120). There, we discuss the difference between LCNs and original Residual Networks,  Deep Supervision, DenseNets and DenseFormers,  which are models referenced in our Related Work. The key distinction is that in all these models each layer is connected to all preceding layers (for Residual and Deep Supervision check [1] eq.1 and also [2], for DenseNet see in [1] eq.2 and for DenseFormer see in [3] Fig. 1 and equations in page 3), whereas LCNs possess a fundamentally different structure, connecting each layer only to the output, as demonstrated in Equations 1–4 and detailed in Appendices A and G.
>
> [1] https://arxiv.org/abs/1608.06993, [2] https://arxiv.org/abs/1605.06431, [3] https://arxiv.org/abs/2402.02622
>
> - Quantitative comparison of LCNs and Related models
>
> We quantitatively compared LCNs with Residual Networks across various settings in the main paper and also showed that Deep Supervision does not exhibit the same behavior (Appendix C).
>
> 2. “the link to biology and the brain is interesting, but after the initial abstract it is hardly addressed. The authors report longer training times as more akin to 'biological brain networks', but I'm not sure which result even supports this (in Fig 8 it trains faster?). That LCNs handle noise better is interesting, but an intuitive explanation/link to previous literature as to why this occurs would be very helpful in its interpretation.”
>
> We revised the paper to better highlight the connections between LCNs and biological brain networks. Specifically we highlighted the connection between LCNs and layer-wise training (Appendix G4), a method that is employed in NN training and is motivated by cognitive neural development in the prefrontal cortex [1]. We further detailed the macroscopic similarities of LCNs and biological NN in our conclusions. We also briefly note that studies have indicated layer V in the neocortex receives inputs from all other neocortical layers and functions as the primary output layer [2,3], much like the output layer of an LCN.
>
> We direct the Reviewer to reference [4], which demonstrates that architectures interpolating between residual and feedforward networks exhibit enhanced robustness to noise, as compared to standard residual networks (intuitively skip connections transfer input noise to the entire network).
>
> In the majority of the experiments (Mixer on Cifar-10 Sec 3.4 footnote 9, ViT on ImageNet Sec. 3.2, BERT pre-training Appendix F) LCNs required longer training times compared to residual baselines.
>
>  [1] https://www.frontiersin.org/journals/neuroanatomy/articles/10.3389/fnana.2011.00029/full
> [2] https://www.frontiersin.org/journals/cellular-neuroscience/articles/10.3389/fncel.2015.00233/full
> [3]https://www.frontiersin.org/journals/neuroscience/articles/10.3389/neuro.01.1.1.002.2007/full
>
> [4] https://arxiv.org/abs/2006.05749
>
> 3. “It appears from the figures that all results are derived using a single random initilisation seed? If so, I would strongly recommend re-running these analyses over several seeds, otherwise it is very difficult to draw any strong conclusions.”
>
> We added in the manuscript the results from multiple runs (Table 1).
>
> 4. “Some of the segways between sections could be smoother. For example short introductions to sections 3.1/7 would help. Some figures could also be clearer. For example a few instances where the axis labels were confusing. E.g. what are the axis labels in Fig 2? I presume w and the error gradient with respect to w? in fig 3 is weight of hidden1 = w1, and wouldn't it be interesting to look at w2/w3 as well to check if they are sparser for LCN, and maybe absolute values instead of raw values to better show sparsity? In Fig 7 I would prefer an explicit mathematical term relating to an equation than 'std' and 'noise_level'. In section 3.1 a schematic (even if for the appendix) would be helpful to the reader”
>
> We included a short introduction at the beginning of section 3. We also clarified Fig.2 and changed x-axis in Fig.3 and Fig.7 for clarity, as per the reviewer’s suggestion. For section 3.1 we included a reference to Fig.1 to visualize the subnetworks we describe.
>
> - We continue our response in the comment below due to characters limit.

---

> ### Author Response · Authors · 2024-11-26
> **Reply to Reviewer Zpq5 (2/2)**
>
> Answering Reviewer's questions :
>
> - “line 54: 'Contemporary deep learning architectures aim for    overparameterization' - what does this mean?”
>
> We direct the Reviewer to references [1–5].  Modern neural networks are overparameterized, meaning they contain a significantly larger number of parameters than required to fit the training data. This design exhibits intriguing properties and has been the subject of extensive analysis.
>
> [1] https://arxiv.org/abs/1912.02292, [2] https://arxiv.org/abs/1812.11118, [3] https://arxiv.org/abs/1811.04918, [4]https://arxiv.org/abs/2206.04569, [5] https://arxiv.org/abs/2311.14646
>
> - “is the residual network given by equation (3) a specific type of residual network? It is introduced there rather abruptly for a non-expert reader”
>
> We added the equation of the residual network  in Appendix A. Since in 2.1 we perform a simple mathematical analysis we are considering a residual network given by the equation:
>
> $$ x_{l+1} = f_{l+1}(x_l) + x_l = w_{l+1}x_l + x_l$$
>
> - “for the toy example the models are trained for 500 epochs (1000 examples per epoch). This seems like a lot for such a simple problem?”
>
> We trained with simple SGD and we used a small learning rate (=0.001), leading to  slow convergence. Re-running the experiment with larger lr (=0.01) we get similar results with 100-150 epochs.
>
> - “line 259: 'possible indication of overfitting' - is Fig 5 showing the training or validation accuracy?”
>
> Fig.5 is showing the validation accuracy. In Fig. 5c the intermediate layers of the residual Transformer exhibit an irregular non-smooth behavior (compared to 5a and 5b of the same model)- that is why we think it might be an indication of overfitting.
>
> - “typos: line 210: brackets around citation line 326: -> architectures line 411: -> directly paragraph 460: brackets around (multiple) citations line 492: -> effectively”
>
> We revised the manuscript and corrected the mentioned issues.

---

> > ### Comment · Reviewer_Zpq5 · 2024-11-29
> >
> > Thank you authors for your detailed reply.
> >
> > I appreciate the extra effort into improving the link with biology. However, my first concern - that there is not a sufficient comparison to other relevant models - remains.
> >
> > I see now that there is a quantitative comparison with residual connections (e.g. Fig 8). I must say I'm confused as to whether 'vanilla' also means (standard) residual connections, or just a vanilla feedforward network without residual connections (how I initially interpreted it), as labeled in Fig 5. In any case I would encourage to make this clearer to the reader.
> >
> > Still, from my judgement the primary motivation of this model is reduced parameterization in the form of synaptic pruning, and for this reason there should be at least some quantitative comparison with other (ideally SOTA) pruning methods. As per Chen et al. 2024 there exists a large number of pruning methods which can be used during training or, as in the case of LCN, after training. Demonstrating the benefits of LCN compared to some of these methods - e.g. final network performance + # parameters required - is in my view a minimum requirement. I apologise for not making this point - comparison with other pruning techniques specifically - clear in my original review.
> >
> > Overall, my score remains unchanged. I appreciate that the paper 'ultimately introduces a new architecture' as per discussion with reviewer Fzkp, but a new architecture is itself only as valuable as the specific problems it solves/improves upon against existing models.

---

> > > ### Author Response · Authors · 2024-11-30
> > > **Reply to Reviewer Zpq5**
> > >
> > > We thank the reviewer for the feedback.
> > >
> > > We appreciate the reviewer's concern about comparisons with pruning methods. However, we believe this stems from a fundamental misunderstanding of our contribution. Traditional pruning methods are inherently lossy—they achieve parameter reduction by accepting some performance degradation. In contrast, LCNs achieve equivalent or better (under noisy input or data sparsity) performance while naturally using fewer layers (30–80% reduction) through architectural design alone. This lossless reduction in network size, achieved without any performance trade-off, represents a qualitatively different contribution.
> > >
> > > LCNs represent a novel architectural approach, not just a method for pruning. Existing pruning methods can still be seamlessly integrated with LCNs to potentially achieve even greater efficiency, as they do with traditional neural architectures. However, comparing LCNs directly with pruning methods would be like comparing lossless vs. lossy compression—they serve fundamentally different purposes and address different problems.
> > >
> > > Our extensive experiments across various architectures and tasks demonstrate that LCNs match the performance of traditional neural network architectures (in our experiments, residual networks) while automatically detecting and eliminating redundant layers through their training dynamics, without requiring additional criteria or regularizers. We believe that this represents a significant advancement in neural network design, paving the way for a new direction of more dynamic architectures where depth is no longer a fixed hyperparameter but can be dynamically adapted by the architecture itself to suit the task at hand.

---

> > > > ### Comment · Reviewer_Zpq5 · 2024-12-02
> > > >
> > > > Dear authors,
> > > >
> > > > What fundementally concerns me is that this 'lossless' reduction depends on highly redundant original model with a significant number of (as you identify unnecessary) layers. I do not agree with the concept that ML researchers 'aim for overparameterization', I believe in ML the number of parameters if purely based on emperical performance.
> > > >
> > > > What would impress me is if you were able to demonstrate such performance gains starting with an original model with the same number of layers/parameters as, e.g. the baseline models in Table 4 in https://arxiv.org/pdf/2403.17921. That would make it much easier for me to directly compare your method to 'lossy' methods. I.e. can LCN be succesfully applied to realistic network architectures, or only those with a tremendous amount of original layers?
> > > >
> > > > I am also rather cynical that LCN can be branded as 'lossless' versus alternative 'lossy' pruning methods. This must surely depend on the original network architecture/redundancy as per above and the type of task.

---

> > > > > ### Author Response · Authors · 2024-12-02
> > > > > **Reply to Reviewer Zpq5**
> > > > >
> > > > > Thank you for your timely reply. We address your concerns next:
> > > > >
> > > > > Architectural Redundancy vs. Pruning: Your concern suggests that LCNs require starting with highly redundant models, which is not the case. We use the exact same architecture that achieves best performance for  traditional models, e.g., ViT-Base with 12 layers (86M params) from your example. Our approach demonstrates that this standard architecture inherently contains redundancy, as only 6-7 layers are needed for the same level of performance. This isn't about starting with unnecessary layers, but revealing the intrinsic redundancy present in existing network designs. Of course, we are not the first to suggest that redundancy can exist in neural networks, e.g., see [https://arxiv.org/pdf/1610.01644, https://arxiv.org/pdf/1605.06431],  we just propose a novel way to reduce such redundancies.
> > > > >
> > > > > Further, we want to stress that LCNs employ an adaptive computational strategy, not a static pruning method. Unlike traditional pruning, which imposes external constraints, LCNs adapt to task complexity organically via its architecture. When full network capacity is required, all layers are used; when fewer layers suffice, the network naturally converges to a more efficient configuration. This is evidenced both in Appendix G and Figure 6 of our paper. The distinction is similar to that between dropout and L2 regularization: while L2 explicitly penalizes weights (like pruning explicitly reduces parameters), dropout adaptively manages network capacity (like LCNs organically adapt layer usage).
> > > > >
> > > > > We hope this clarification addresses your concerns and highlights the novel contribution of LCNs. While we haven't explored every possible configuration and training scenario, our extensive experiments across different architectures and tasks demonstrate the effectiveness of LCNs. We look forward to seeing how the community builds upon these findings.

---

> > > > > > ### Comment · Reviewer_Zpq5 · 2024-12-02
> > > > > >
> > > > > > Thank you authors.
> > > > > >
> > > > > > Ok, I'm glad to understand that the original ViT-Base does use 12 layers itself, and this does strengthen the results by making it more comparable to existing work. I will raise my score by 1.
> > > > > >
> > > > > > I am still concerned however by the different numbers reported versus other studies. For example, if the ViT-Base has the same network architecture, why does it achieve ~5% less accuracy on the ImageNet task versus that reported in https://arxiv.org/pdf/2403.17921?
> > > > > >
> > > > > > Also, in Table 1 I notice the accuracy for LCN is slightly less on the Amazon task versus vanilla. This is not by itself concerning and is indeed within the margin of error, but still reinforces my belief it's dangerous to call LCN 'lossless'.

---

> ### Author Response · Authors · 2024-12-03
> **Reply to Reviewer Zpq5**
>
> Thank you for raising your score and for these additional points.
>
> Regarding ImageNet accuracy: The difference in reported accuracies is  due to batch size and associated training settings. We use a batch size of 256 due to computational constraints, while the original  ViT paper uses 4096; all other hyper-parameters are identical. This impact of batch size is well-documented - for example, the same ViT-Base implementation achieves 74.6% with the original training setup versus 72.6% with batch size 1024 (see https://github.com/ssuncheol/vision-transformer-pytorch). Our results are consistent with these findings. Note that this performance gap could be decreased with extensive hyperparameter tuning, but most importantly, we compare LCN and vanilla (residual) architectures under identical training conditions, ensuring a fair comparison of the architectural contribution. We will add a note about training conditions vs performance in the final paper for clarity.
>
> On performance differences: The small variations in performance between LCNs and traditional architectures can be explained by their different generalization characteristics. LCNs might show slightly lower performance when a model is overfitting to a dataset (as the additional capacity of traditional architectures can help memorize the training data), but tend to outperform when better generalization is needed. This is consistent with their adaptive layer usage and explains the small variations you note in Table 1.
> We use "lossless" to convey that LCNs achieve comparable performance while automatically using fewer layers, but we're happy to use "performance-preserving" if you feel it would be clearer.

---

### Official Review · Reviewer_qQay · 2024-11-03

**Soundness:** 2
**Presentation:** 3
**Contribution:** 2
**Rating:** 5
**Confidence:** 3

**Summary:**

The authors add long-range connections to standard transformers and show that this enables effective pruning, perhaps like that in the real brain.

**Strengths:**

This is an interesting and creative idea.  It is also true that it is supported by the neuroscience literature is a reasonable possibility.

**Weaknesses:**

The evaluations are too simple.  Like many things in machine learning, something that works for simple versions of tasks might break down for larger tasks.   I am most knowledgeable about vision.  I would like to see similar results for imagenet rather than just CIFAR-10 to be really sure that these results are actually going to be impactful.  (Apologies if I missed it if this was included.) I am not competent to juge the meaningfulness of the language results.

**Questions:**

Can you illustrate results on a stronger task like imagenet? Or perhaps CIFAR-100?

---

> ### Author Response · Authors · 2024-11-26
> **Reply to Reviewer qQay**
>
> We thank the Reviewer for taking the time to examine our work and for providing valuable insights and comments. We will now address the questions raised:
>
> “The evaluations are too simple. Like many things in machine learning, something that works for simple versions of tasks might break down for larger tasks. I am most knowledgeable about vision. I would like to see similar results for imagenet rather than just CIFAR-10 to be really sure that these results are actually going to be impactful. (Apologies if I missed it if this was included.) I am not competent to juge the meaningfulness of the language results.”
>
> We presented our proposed architecture and validated it through both theoretical analysis (Appendix G) and experimental studies across text and vision modalities. Our experiments included simpler datasets such as IMDb (Fig. 5), Amazon (Fig. 4b), and CIFAR-10 (Fig. 6). Additionally, in the original submission we evaluated LCNs on ImageNet (Fig. 4b, Fig. 7, Table 1). We believe this aligns with the reviewer’s suggestion.
>
> Furthermore, in the revised version we include another large-scale experiment on BERT pre-training and fine-tuning (see Appendix F).
> We note that across all these experiments, LCNs were trained efficiently, delivering performance on par with their residual network counterparts and showing improved robustness against noise,  while consistently demonstrating their ability to prune redundant layers, achieving a 30-80% reduction in the number of layers compared to baseline models.

---

> > ### Author Response · Authors · 2024-12-01
> >
> > We appreciate that you are likely to review multiple other papers, however, as we approach the end of the discussion period, we would greatly appreciate your feedback on our rebuttals. We are happy to address any remaining concerns or questions to improve the paper further. Thank you so much!

---

### Official Review · Reviewer_RKnz · 2024-11-04

**Soundness:** 2
**Presentation:** 3
**Contribution:** 2
**Rating:** 6
**Confidence:** 4

**Summary:**

Inspired by brain synaptic pruning, they proposed long connection neural networks that eliminate top layers without sacrificing performance, resulting in a significant reduction in inference time.

**Strengths:**

1. This work accelerates inference and reduces memory usage without sacrificing performance.
2. This architecture can be used complementarily with pruning or dynamic inference techniques.

**Weaknesses:**

1. The validity of the structure is verified only on the base ViT, but there is a lack of experimental and theoretical validation for larger model structures.
2. If the author claims that this is a fine-tuning method, the time spent during model training is typically included in the overall time assessment. Fine-tuning often involves adjusting a pre-trained model on a specific dataset, and this process can require significant computational resources and time. However, if the focus is on inference speed or deployment efficiency, the discussion (time, GPU memory and other computation cost) might be reported during the fine-tune process .
3. It sounds like you' re noting that the determination of the final model layer relies heavily on experimental validation, which can indeed be resource-intensive and time-consuming. This approach can lead to significant computational costs, especially when iterating through different configurations to find the optimal structure.
4. While verifying the effectiveness of a structural design on a classification task provides important insights, it doesn't necessarily guarantee that the same design will perform well across different tasks, such as object detection, segmentation, or generative modeling. To better justify the generalization to other tasks, the authors could provide theoretical justifications or include empirical evidence.

I am glad to raise my score if the authors can address the above points.

**Questions:**

See weaknesses.

---

> ### Author Response · Authors · 2024-11-26
> **Reply to Reviewer RKnz**
>
> We thank the Reviewer for taking the time to examine our work and for providing valuable insights and comments. We have taken them into account resulting in a much improved manuscript.
>
> 1. “The validity of the structure is verified only on the base ViT, but there is a lack of experimental and theoretical validation for larger model structures.”
>
> - theoretical validation
>
> We provide, in Appendix G, a comprehensive theoretical analysis of the training dynamics of LCNs based on gradient flows for different layers. We also validate these theoretical claims experimentally by visualizing the gradient dynamics of LCNs during training (see Appendix G.4).
>
> - experimental validation
>
> We provide further evidence, in Appendix F,  on the broad applicability of the proposed architecture by running a pre-training experiment on BERT (followed by fine-tuning on three tasks). The results are consistent with the rest of the training recipes in the paper.
>
> 2. “If the author claims that this is a fine-tuning method, the time spent during model training is typically included in the overall time assessment. Fine-tuning often involves adjusting a pre-trained model on a specific dataset, and this process can require significant computational resources and time. However, if the focus is on inference speed or deployment efficiency, the discussion (time, GPU memory and other computation cost) might be reported during the fine-tune process .”
>
> All experiments in the main body of the paper consist of models trained from scratch. We clarified that the reported metrics and results related to inference speed and memory usage pertain to the training process.  Moreover, to show the general applicability of our architecture we also experimented on fine-tuning (Appendix D) as well as pre-training+fine-tuning (Appendix F). In all cases we provide  details on the training and evaluation protocol we followed (computational resources, epochs).  We note that:
>
> Our experiments show that while LCNs require ~2x longer training time (700 vs 300 epochs for ViT), they achieve:
> - 40% faster inference (13.9ms → 8.3ms)
> - 41% reduction in parameters (86M → 51M)
> - 41% smaller storage size (330MB → 195MB)
>
> Importantly, the longer training is a one-time cost that enables permanent inference benefits. The pruned architecture can be distributed for deployment without requiring end users to perform the longer training"
>
> 3. “It sounds like you' re noting that the determination of the final model layer relies heavily on experimental validation, which can indeed be resource-intensive and time-consuming. This approach can lead to significant computational costs, especially when iterating through different configurations to find the optimal structure.”
>
> For determining the final model’s number of layers we indeed use a validation set. However, this set is small (5% of the full dataset in our ImageNet experiment), allowing for quick validation. Our method requires only a single pass per batch to compute the output of each layer (as detailed in Section 3.1), and a single evaluation is sufficient, without the need for multiple iterations (as opposed to methods proposed in [1], [2]). We are also working on super-light algorithms for determining a sample-specific layer depth (see Future Work).
>
> [1]https://proceedings.neurips.cc/paper_files/paper/2022/file/3b11c5cc84b6da2838db348b37dbd1a2-Paper-Conference.pdf
>
> [2] https://arxiv.org/pdf/2402.11187
>
>
> 4. “While verifying the effectiveness of a structural design on a classification task provides important insights, it doesn't necessarily guarantee that the same design will perform well across different tasks, such as object detection, segmentation, or generative modeling. To better justify the generalization to other tasks, the authors could provide theoretical justifications or include empirical evidence.”
>
> We present some evidence supporting generalization:
> - Appendix F demonstrates successful application to masked language modeling and transfer learning (BERT).
> The results are consistent with our observations when training LCNs from scratch, delivering performance on par with the residual baselines while utilizing 25% of the full model’s layers.
> - The theoretical analysis in Appendix G shows that the benefits arise from fundamental gradient flow properties that are task-independent. The analysis is performed for an arbitrary task modeled by a loss function $L$ that we aim to minimize and it is further supported by empirical evidence.
> - Empirically, we show robust transfer learning capabilities from ImageNet to CIFAR-10 (Appendix D).
>
> Nonetheless, we agree more validation on diverse tasks would be valuable and have explicitly noted this as a limitation in Section 9.
> We are currently working on scaling up our experiments to billion-parameter LLMs, as discussed in our Future Work section.

---

> > ### Comment · Reviewer_RKnz · 2024-11-26
> >
> > I thank the authors for their detailed replies. I have raised my score accordingly.

---

> > > ### Author Response · Authors · 2024-11-30
> > > **Reply to Reviewer RKnz**
> > >
> > > We thank the Reviewer for the valuable feedback and the updated score.

---

### Author Response · Authors · 2024-11-26

We thank the Reviewers for their rigorous and productive feedback, which has substantially improved our manuscript. We have addressed their recommendations and outlined both the major changes and our responses to individual comments. Below, we detail the specific improvements made to the manuscript's technical content and presentation:

-  Included in Appendix G a more comprehensive theoretical analysis of the training dynamics of LCN, as requested by R1 (RKnz), R2 (Qqay) and R4 (Fzkp). The analysis provides an explanation for the “layerwise-training” behavior observed in the LCN experiments and why this behavior generalizes to different tasks and training regimes.

-  To further demonstrate the general applicability of the proposed LCN architecture we added a pretraining experiment on BERT as suggested by R1 (RKnz) and implied by R3 (Zpq5) (Appendix F).

- Revised the manuscript for minor errors, enhanced the introductions, clarified the figures as pointed out by R3 (Zpq5).

-  Added multiple run results (Table 1) as requested by R3 (Zpq5).

-  Enhanced the literature review as requested by R3 (Zpq5).

-  Presented some state-of-the-art pruning techniques applied to tasks and models similar to ours, and compared them with our method as requested by R4 (Fzkp).

---

### Author Response · Authors · 2024-12-02

As the discussion period draws to a close, we would like to express our sincere appreciation to all reviewers. Your thoughtful comments have helped us better understand and position our work. The review process has motivated major breakthroughs in our theoretical analysis of derivative flow, led us to fundamentally rethink our contribution of introducing overconnectivity/sparsity as an architectural choice rather than an engineering trick, and opened exciting avenues for future work.
Given that today (Dec 2) is the final day of requesting input from authors, we welcome any additional questions or requests for clarification

---

### Author Response · Authors · 2024-12-03

As the discussion period concludes, we'd like to highlight the consistency of our findings across a wide range of scenarios. LCNs have demonstrated consistent behavior across:
* Multiple architectures (MLPs, Transformers, Mixers)
* Various tasks (classification, language modeling)
* Different training paradigms (from-scratch training, fine-tuning, pre-training)
* Different model scales and layer configurations

These results strongly suggests that the benefits of LCNs are fundamental to the architectural design rather than specific to particular configurations. However, we understand that we provide limited experimental proof for larger top-performing models. We are actively working on two significant extensions:
1. A large-scale (1B parameter) LLM pre-training and fine-tuning experiment to extend our preliminary BERT findings
2. The ViT-ImageNet experiment with 4096 batch size to match standard training configurations

With the reviewers' and Area Chairs' permission, we would include these results (when available) in an appendix of the final paper, as they would provide additional validation of our findings at larger scales.

---

### Meta-Review · Area_Chair_BKya · 2024-12-20

**Metareview:**

This paper presents an approach for reducing the compute required for inference in deep neural networks. The authors claim that when one adds long-distance residual connections to a deep neural network (i.e. connections from lower layers straight to the final layer), then a lot of the computation gets put on earlier layers and the later layers can be dropped from inference without any reduction in accuracy. To support this claim they present empirical studies on classification tasks for a few different network architectures showing that they can maintain accuracy even after pruning away later layers from these long-range connection networks.

The strengths of this paper are that it is well-written and novel, and presents a potentially useful technique for improved efficiency at inference. The weaknesses are that the purported links to biology are tenuous at best, and the authors’ provide relatively little evidence that this technique is better than other techniques for improved efficiency, e.g. either pruning or conditional computation (both of which can achieve similar improvements in inference cost).

Based on these considerations, the AC feels that this paper, though potentially an important contribution, requires more work before it is ready for publication at a major ML conference. In particular, though the authors said in rebuttal that they didn’t compare to other pruning techniques because they are lossy, it seems pretty reasonable to ask how much is the comparative gain in efficiency and *how* bad pruning techniques are in comparison to this one? That way, readers can judge whether they would be willing to take a small hit in accuracy for even better efficiency, if that was indeed the trade-off. As noted by the reviewers, the architectures and tasks used could also be expanded. And, there doesn’t seem to be any comparison to other conditional computation techniques (see e.g. https://arxiv.org/abs/2404.02258), and in the AC’s assessment, this is a key metric of performance to judge this technique. Finally, the AC would encourage the authors to consider either toning down or removing the claimed links to biology as the results here don’t really have any clear connection to how pruning works in the brain (which is not generally a process of removing entire layers regions from the network at inference, but rather, selective removal of synapses across the entire hierarchy).

**Additional Comments On Reviewer Discussion:**

The discussion was overall courteous and productive. The reviewers raised several points, and there was clear agreement on the limited evidence and lack of comparisons to relevant existing techniques. The authors attempted to address these concerns, and did partially, leading some scores to be raised. But, ultimately, they did not manage to convince the reviewers enough to see the scores raise into a territory of clear accept, and so, the paper remained a borderline case that the AC had to make a judgement call on. In the end, the AC was convinced that the reviewers were correct that the paper was still missing some key experiments.

---

### Decision · Program_Chairs · 2025-01-22

Reject